# Informed and Empowered: A Pre–Post Evaluation of a Whiteboard Video for Sexual Health Education in Female Adolescents and Young Adults with Cancer

**DOI:** 10.3390/curroncol32120681

**Published:** 2025-12-01

**Authors:** Natalie Pitch, Anjali Sachdeva, Jennifer Catsburg, Mackenzie Noyes, Sheila Gandhi, Rebecca Côté, Chana Korenblum, Jonathan Avery, Abha A. Gupta

**Affiliations:** 1Department of Pediatrics, University of Toronto, Toronto, ON M5G 1X8, Canada; chana.korenblum@sickkids.ca (C.K.); abha.gupta@sickkids.ca (A.A.G.); 2Division of Pediatrics, The Hospital for Sick Children, Toronto, ON M5G 1X8, Canada; 3Temerty Faculty of Medicine, University of Toronto, Toronto, ON M5S 1A8, Canada; anjali.sachdeva@mail.utoronto.ca; 4Adolescent and Young Adult Program, Princess Margaret Cancer Centre, University Health Network, Toronto, ON M5G 2C4, Canada; jennifer.catsburg@uhn.ca; 5Faculty of Health, University of Waterloo, Waterloo, ON N2L 3G1, Canada; mackenzie.noyes@uhn.ca; 6Division of Oncology, The Hospital for Sick Children, Toronto, ON M5G 1X8, Canada; sheila.gandhi@sickkids.ca; 7Teen Cancer Connection Program, The Hospital for Sick Children, Toronto, ON M5G 1X8, Canada; rebecca.cote@sickkids.ca; 8BC Cancer, Provincial Programs, Supportive Care, Vancouver, BC V5Z 4E6, Canada; jonathan.avery@bccancer.bc.ca; 9School of Leadership Studies, Royal Roads University, Victoria, BC V9B 5Y2, Canada

**Keywords:** adolescents and young adults, cancer survivorship, sexual health education, whiteboard video, patient education, oncology, females

## Abstract

Sexual health is important but often overlooked for female adolescents and young adults (AYAs) with cancer, despite its strong impact on quality of life. We created a short whiteboard animation video to give clear, practical information on how cancer and its treatment can affect sexual health and offer strategies for managing these changes. Ninety AYAs across Canada watched the video and completed surveys before and after. Knowledge scores improved by nearly 20%, with greater gains among participants with a high school education or less, and younger participants tended to show larger improvements in sexual health knowledge after watching the video. Most described the video as easy to follow, helpful, and a resource they would recommend to others, while suggesting improvements, including shorter length, enhanced visuals, and more age- or relationship-specific content. These findings suggest that whiteboard videos can support sexual health education and help normalize these conversations in cancer care.

## 1. Introduction

When adolescents and young adults (AYAs)—defined in North America as individuals between the ages of 15 and 39 years—are diagnosed with cancer, their quality of life is distinctly affected due to the intersection of developmental, psychosocial, and biological factors [1]. Within this context, sexual health emerges as a critical, yet often underappreciated, determinant of quality of life [2,3]. Sexual health encompasses one’s sexual functioning, as well as their subjective experience of sex, which may be impacted by factors including body image, self-esteem, and relationship satisfaction [4]. Cancer and its treatment can affect all these dimensions, particularly in the AYA population [5].

Sexual dysfunction and related challenges are common in both male and female AYAs with cancer; however, females report a higher burden of sexual health problems compared to males [6,7]. Common issues include decreased libido, difficulty achieving orgasm, and pain with intercourse, arising from an interplay of physiological, social, and psychological factors [8,9]. Despite this, AYAs with cancer consistently describe unmet informational needs and a desire to learn information about sexual-health-related changes and management strategies [10,11]. As such, the onus lies with healthcare providers to initiate assessments and offer appropriate care for cancer-related sexual dysfunction resulting from cancer and its treatment.

One-on-one clinical discussions can be valuable but are inconsistently offered; clinicians report limited time, inadequate training, and discomfort as common barriers to initiating sexual-health conversations with AYAs [12,13,14]. Given these barriers, clinicians emphasize the need for supportive, developmentally appropriate materials that can aid in fostering sexual-health conversations with AYAs [4,15,16,17]. This gap has important implications, as educational tools can support more informed, comprehensive, and proactive engagement in sexual healthcare.

However, many cancer education materials still contain complex conceptual information presented in dense, text-heavy formats, which can be challenging for the average reader to understand [18,19]. Moreover, our own environmental scan of existing sexual-health resources relevant to female AYAs (pamphlets, websites, podcasts and videos), revealed that resources labeled as AYA-focused combined fertility and sexual-health information and were predominantly text-based, with only a small number incorporating video or visual formats. Among the resources that do exist, to our knowledge, no peer-reviewed study has evaluated the effectiveness of a sexual-health education tool specifically for female AYAs with cancer.

Whiteboard animation videos are emerging patient education tools that have shown promise in addressing these gaps. These videos use simple sketches drawn in real time on a blank background, paired with narration. Whiteboard videos have shown to be effective in improving knowledge, engagement and recall across various health topics, including oncology [20,21]. Their straightforward, conversational style makes them particularly well-suited for conveying sensitive or complex information to the AYA population [21,22]. To address the gap in sexual health education for female AYA patients with cancer, the AYA Program at the Princess Margaret (PM) Cancer Center developed a 13 min whiteboard video with visual sketches to explain the importance of sexual health, the impact of cancer treatment, and strategies for managing common concerns. The video is intended to complement clinical discussions and enhance patient-provider communication.

The primary aims of this study were to evaluate the effectiveness of a whiteboard video in improving sexual health knowledge and to assess participants’ overall impressions of the video. Secondary aims were to examine its understandability, actionability, and readability, as well as participants’ general interest in sexual health and prior experiences discussing it.

## 2. Materials and Methods

### 2.1. Intervention Development: Video Creation

The AYA Program at the PM Cancer Center conducted an environmental scan of available sexual-health resources for AYA cancer populations. This scan reinforced the need for a more succinct, visually engaging, and developmentally appropriate tool tailored to oncology care. In response, a script and video were created by a medical student with the input of multiple oncologists, a pediatrician, a nurse practitioner, a clinical nurse specialist, two pelvic floor physiotherapists, and a social worker. The script was reviewed by a patient education specialist who made recommendations to make the video more inclusive. The content of the video was grounded in current evidence-based information to ensure clinical accuracy and relevance (see Appendix A). A team of medical illustrators created the animation, and a professional voice actor read the script. 

The understandability and actionability of the video were evaluated using the Patient Education Materials Assessment Tool for Audiovisual Materials (PEMAT-A/V). The PEMAT-A/V is a validated tool designed to determine whether patients can understand and act on information provided [23]. Health literacy is well recognized in the literature as a challenge for patients affecting informed medical decision making [24]. Six reviewers uninvolved in the creation of the video independently evaluated the video using the PEMAT-A/V tool. Understandability and actionability scores of 80% or greater are considered acceptable. Readability was evaluated using six free, online calculators (Flesch Reading Ease, Gunning Fog, Flesch-Kincaid Grade Level, Coleman-Liau Index, SMOG Index, and Linsear Write Formula). All six were reported to reflect variability across tools, yielding a range from Grade 10 to college level. Because the analysis was descriptive, no single calculator or predefined threshold was applied.

### 2.2. Questionnaire Development

To our knowledge, there are no validated tools specific to measuring knowledge on sexual health in cancer in the AYA cancer population. Thus, we developed a questionnaire to assess patient interest in sexual health, knowledge of sexual health before and after watching the video, as well as satisfaction and impressions of the video. The questionnaire included a combination of multiple-choice, open-ended questions, and Likert scales. It comprised four sections: (1) sociodemographic information, (2) interest/previous experience with sexual health discussions, (3) knowledge about sexual health and (4) impressions of the video.

The nine knowledge items were derived directly from the educational content presented in the video and mapped to the three stated learning objectives: identifying common sexual health changes, learning strategies to manage these changes, and increasing comfort discussing sexual health with healthcare providers. Examples include: (1) “After receiving chemotherapy, how long should you wait until it is safe to kiss your partner?”, (2) “Is it always have safe to have sex during chemotherapy?”, and (3) “Which of the following can help manage vaginal dryness?” A small pilot was conducted with two eligible AYA participants (not included in the final sample) to assess comprehensibility and completion time. To enhance content validity, the draft questionnaire was reviewed by a pediatric oncologist, an adolescent medicine physician, a social worker, and a clinical nurse specialist, who provided feedback on clarity, appropriateness, and clinical relevance.

### 2.3. Sampling and Recruitment

Participants were recruited across Canada from June 2024 to March 2025. Inclusion criteria included (1) current age 15–39 (2) assigned female at birth (3) English speaking and (4) at least 1 month post-diagnosis up to 10 years post-treatment. 

Recruitment occurred through several methods. Patients were identified by primary teams at SickKids and PM Cancer Center and agreed to receive the study link. Study recruitment was extended nationally through social media (PM AYA Instagram account and website) as well as an invitation letter to the Canadian Cancer Society, Young Adult Cancer Canada, AYA CAN, ANew Research Collaborative and Wellspring. The letter described the study, provided a link to the online questionnaire, and contact information to answer questions. Given that recruitment used a multi-faceted approach involving clinic staff, social media, and partner organizations, and participation was anonymous, the number of individuals approached through each source could not be tracked.

Participants aged 15–18 years who completed the survey received a gift card, whereas participants aged 19–39 years were entered into a draw for three gift cards. This approach was chosen to ensure adequate engagement of adolescents, who may require stronger incentives to participate in research.

### 2.4. Data Collection

Research Electronic Data Capture (REDCap) version 15.7.6 (Vanderbilt University, Nashville, TN, USA) was used to disseminate the questionnaires along with the video. Each participant received a participant identification linking their demographic information to their survey responses. Participant responses to this survey were anonymous. The survey included five sections: (1) Consent, (2) Demographics, (3) Pre-video questionnaire, (4) Video and (5) Post-video questionnaire. This study was approved by the SickKids (1000081417, 5 April 2024 and Princess Margaret Hospital (23-6062, 16 April 2024) Research Ethics Board. Informed consent was obtained electronically; participants reviewed a study information page and consent was implied by survey completion, with the option to withdraw at any time by closing the survey.

### 2.5. Statistical Analysis

The PEMAT-A/V is derived by taking an average of 14 items for understandability and four items for actionability (see Appendix A). Six female AYA raters evaluated the video using the PEMAT-A/V tool to assess understandability and actionability. Two raters were oncology nurses familiar with patient education, and four were non-healthcare participants representing the target population. All raters reviewed the PEMAT-A/V user manual to ensure consistent interpretation of the scoring criteria. Ratings were completed independently without discussion or consensus review. Item scores were averaged to determine overall understandability and actionability scores. Inter-rater reliability was assessed using a two-way random-effects, average-measures model (ICC [2,k]).

The readability score was averaged across the six calculators. Using Microsoft Excel version 16.103.1 (Redmond, WA, USA), participant characteristics were summarized using descriptive statistics and continuous data were summarized with means and standard deviations. Likert-scale responses were summarized using proportions of participants who selected favorable response options (e.g., ‘agree’ or ‘strongly agree’). Knowledge scores were calculated as the percentage of correct responses out of nine multiple-choice items (range 0–100%). Each correct response was scored as 1, while incorrect answers, “I don’t know,” and “prefer not to answer” responses were scored as 0.

A paired-samples t-test was conducted to evaluate the change in knowledge following the video. To assess predictors of knowledge score change, a multiple linear regression analysis (N = 90) was performed with age, education level, and treatment status as independent variables. The primary endpoint was the change in overall knowledge score from pre- to post-video. No adjustment for multiple comparisons was applied, as secondary analyses were exploratory in nature. Education was recoded based on the highest level of education selected, categorized as high school or less, college/university/technical school, or post-graduate education. Treatment status was coded as currently receiving treatment (yes vs. no). Additionally, a Pearson correlation was performed to examine the bivariate relationship between age and knowledge score change. Model assumptions were reviewed, and no evidence of non-normal residuals or multicollinearity was identified.

In a separate analysis, logistic regression (N = 81) was used to identify predictors of whether participants had previously discussed sexual health with a healthcare provider. The outcome variable was binary (yes vs. no), and predictors included age, education level, treatment status, and perceived importance of sexual health to overall well-being. Participants who selected “I prefer not to answer” or had missing responses for any of the included variables were excluded from this analysis (N = 9). All analyses were conducted using complete cases.

Three reviewers (N.P., A.S., and A.A.G.) analyzed the qualitative responses independently using inductive coding and recorded suggested codes in a Microsoft Excel spreadsheet. The coding scheme was revised to develop a unified system used to code responses (Appendix A).

## 3. Results

### 3.1. Patient Education Best Practices

The average PEMAT-A/V score was 96% (SD 2%) for understandability and 94% (SD 9%) for actionability. The actionability score was most negatively influenced by low scores on an item stating that “the material provides a tangible tool whenever it could help the user take action” (see Appendix A). Inter-rater reliability demonstrated good to excellent agreement across raters (Understandability: ICC = 0.89, 95% CI 0.63–0.98; Actionability: ICC = 0.76, 95% CI 0.34–0.94). The readability of the script ranged from grade 10 to college level, depending on the scale used. These readability scores were calculated from the video script, which reflects the spoken narration.

### 3.2. Demographics and Patient Characteristics

A total of 90 patients (mean age = 28.3, SD = 8.4) completed the study. Of the 164 online surveys that were started, a total of 91 were completed, rendering a completion rate of 55% (Figure 1). One survey was excluded as the short answers were provided in a non-English language. For those who were approached in person after being deemed appropriate by a member of their primary care team, reasons for declining participation included lack of interest in research participation, not being fluent in English, discomfort with the topic, or feeling that they did not have enough time. The most common diagnoses were breast cancer, leukemia, lymphoma, and sarcoma (Table 1). Eighty (88.9%) participants were from Ontario, which may introduce regional bias and limit generalizability to other provinces. Fifty participants (56%) were actively receiving treatment, 17 (19%) were within their first year following treatment, 17 (19%) were between one year and five years of receiving treatment and six (7%) were between five and 10 years of treatment.

### 3.3. General Knowledge

Patients’ general knowledge on sexual health increased from the pre- to post-video questionnaire by 19.5%, from a mean score of 68.8% to 88.3% after watching the video (*p* < 0.001; 95% CI, 14% to 24%, Cohen’s *d* = 0.89). Item-level analysis showed variability in baseline knowledge across the nine questions. Pre-video accuracy ranged from 27% to 96%, with the lowest scores for items on lubricant safety (27%) and management of vaginal dryness (44%), and the highest for the item about initiating discussions on sexual health with a healthcare provider (96%). Post-video accuracy improved for all items, with the largest gains for the lowest-scoring topics and minimal change for those already well understood (see Appendix A).

The regression model examining predictors of knowledge score change was statistically significant, F(4, 85) = 4.23, *p* = 0.0036, R^2^ = 0.17. Among the predictors, education level was significantly associated with knowledge change: participants with college, university, or technical education had smaller knowledge gains compared to those with high school education or less (B = –17.26, *p* = 0.040). When expressed as standardized coefficients, education demonstrated a moderate association with knowledge change (β = −0.25), while age (β = −0.07) and treatment status (β = 0.06) were not significant predictors (*p* > 0.05). Age and treatment status were not significantly associated with knowledge change (Table 2). However, a Pearson correlation revealed a statistically significant negative correlation between age and knowledge change (r = −0.33, *p* = 0.0016), suggesting that younger participants tended to show greater increases in knowledge at the bivariate level. This discrepancy may reflect overlap between age and education level, as well as limited power to detect small effects in the multivariable model.

### 3.4. Interest/Experience with Discussing Sexual Health

Seventy (78%) participants felt sexual health was important/very important to their overall well-being (Figure 2). Nearly half of participants reported discussing the impact of cancer on their sexual health with a romantic partner (N = 42, 47%), followed by healthcare providers (N = 36, 40%) and friends (N = 23, 26%) (Figure 3). Almost half of the respondents (N = 38, 42%) reported never discussing their sexual health with a healthcare provider; however, 84 (93%) of participants felt it was important/very important to learn about their sexual health. Of those who did discuss sexual health with their healthcare provider, 6 (12%) reported not feeling comfortable and 29 (57%) reported feeling somewhat comfortable. Fifty-four (60%) reported their sexual health concerns were inadequately or not at all addressed by a healthcare provider.

Figure 2: Survey responses regarding perspectives on sexual health communication (N = 90). Bars represent the proportion of participants endorsing each item, with 95% confidence intervals. Most participants reported that it is important to learn about sexual health (93%, 84/90; 95% CI: 88–99%) and that sexual health is important to overall well-being (78%, 70/90; 95% CI: 69–86%). Fewer participants discussed sexual health with a provider (57%, 51/90; 95% CI: 46–67%), and only one-third felt their sexual health concerns were adequately addressed (33%, 30/90; 95% CI: 24–43%).

Figure 3: Survey responses regarding sexual health conversations (N = 90). Bars represent proportions with 95% confidence intervals. Romantic partners were the most common discussion partner (47%, 42/90; 95% CI: 36–57), followed by healthcare providers (40%, 36/90; 95% CI: 30–51), therapists (26%, 23/90; 95% CI: 17–35), and friends (26%, 23/90; 95% CI: 17–35). Few participants discussed sexual health with a caregiver (9%, 8/90; 95% CI: 4–17). Participants could select more than one response.

Logistic regression revealed that age was significantly associated with greater odds of discussing sexual health with a healthcare provider (OR = 1.11, 95% CI [1.01–1.21], *p* = 0.034). Education level, treatment status, and perceived importance of sexual health were not statistically significant predictors.

### 3.5. Satisfaction with Video

Participants were also asked questions about their satisfaction with the video using Likert scales. Seventy-two (80%) participants agreed/strongly agreed that the video represented their ethnic, gender and sexual background. Eighty-five participants (94%) found the video was overall helpful in learning about their sexual health and 80 participants (89%) would recommend this video to a friend with a diagnosis of cancer.

### 3.6. Qualitative Analysis

Analysis of participant responses revealed four key themes (Figure 4). First, participants appreciated the clarity and accessibility of the content. The language was described as non-judgmental and easy to understand, with visuals that reinforced key messages and supported learning. Several participants highlighted that the video effectively explained how cancer can impact sexual health and offered practical strategies for managing these challenges. As one participant shared, “*I really appreciated how clearly the video explained the sexual health challenges I might face during and after my treatment and the solutions that can be used to mitigate those challenges*”.

Figure 4 shows an overview of themes identified in the analysis. Additional illustrative participant quotations corresponding to each theme are provided in the Appendix A.

Second, many participants noted that the video helped to normalize sexual health as a topic of conversation in the clinical setting. Respondents valued the way the video addressed discomfort around these discussions and reassured viewers that it is appropriate and important to speak openly with healthcare providers. One participant noted, “*I appreciated the destigmatization of discussing sexual health with a medical team. It’s easy to feel awkward about it, but the video explained that there is no shame in it.*”

Third, while the video was largely well received, many participants identified important content gaps. The most frequently reported gap was guidance on how to talk with a partner about sexual health concerns during or after cancer treatment. Others suggested the need for more focus on the psychosocial impact of treatment, such as changes in self-image and identity, and more information tailored to specific cancer types. As one respondent shared, “*A conversation on consent, how to balance caregiving and sexual desire, and how to discuss sexual wellness post-treatment with a partner… those discussions need to be had.*”

Fourth, participants suggested improvements to the video. These included shortening the overall length, improving the visual designs, and using a different narrator—preferably a healthcare professional or someone with lived experience. A few younger participants also noted that the content felt less relevant to their age group.

## 4. Discussion

Our study evaluated the effectiveness of a whiteboard education video designed to enhance sexual health knowledge among AYA females with cancer. The video was generally well-received by participants and rated as overall helpful, with the majority reporting that it improved their understanding of sexual health and that they would recommend it to peers. The intervention led to a significant improvement in participants’ knowledge, with an average increase of 19.5% from pre- to post-video assessments.

The use of video-based educational tools in healthcare has promise in improving patient comprehension and engagement, particularly among AYAs. Greenspoon et al. (2020) evaluated a whiteboard video designed to introduce fertility preservation options to female AYAs with cancer [21]. Their study demonstrated that the video effectively increased patient knowledge and received high satisfaction scores, highlighting the potential of this medium to convey complex medical information in an accessible format. This aligns with our findings, as well as broader literature demonstrating that video-based interventions can enhance understanding and influence behaviors across various health topics [25,26]. Notably, younger participants tended to demonstrate greater knowledge gains, which may be partially explained by lower baseline knowledge levels. Participants whose education level was high school or less also showed greater knowledge gains, suggesting that the video was particularly effective in reaching individuals with less prior exposure to the topic. This finding aligns with Greenspoon et al. (2020), who similarly observed greater knowledge improvements in younger AYA participants and proposed that those with less prior exposure to the topic may benefit most from educational interventions [21].

Our video received high scores on the PEMAT-A/V, with an average understandability score of 96% and an actionability score of 94%. These results suggest that the material was well-structured, easy to follow, and delivered in a way that helped viewers identify concrete actions they could take [23]. This is consistent with the design of the video, which presented common sexual health concerns alongside concrete, actionable strategies for managing each issue. While the video scored well on these domains, the average readability level of the script was at a grade 10 to college level, which exceeds the 6th to 8th grade reading level generally recommended for patient-facing educational materials [24,27]. The video would benefit from simplification of the script to better align with recommended readability standards and ensure accessibility for AYA with varying levels of health literacy.

Our findings reinforce the ongoing gap in sexual health communication within AYA cancer care. While the majority of participants (78%) felt that sexual health was important to their overall well-being, nearly half (42%) had never discussed the topic with a healthcare provider. Even among those who had, many reported discomfort or inadequate support. These findings are consistent with existing literature indicating that sexual health is often inadequately addressed in oncology care, despite its recognized importance [7,8,9]. Camejo et al. (2024) reported that only 10.5% of oncology healthcare providers frequently felt prepared to discuss sexual health with patients, while 24.8% almost never have the appropriate tools to address it [28]. Common barriers reported by healthcare providers include lack of training, limited time, and discomfort with the topic [29]. These challenges emphasize the value of accessible, evidence-informed resources such as the educational video developed in this study. Such tools can help educate patients, while also supporting healthcare providers in initiating more proactive and informed conversations about sexual health.

Notably, our study found that older participants were more likely to have previously discussed sexual health with a healthcare provider. This has important implications, as younger AYAs may be at particular risk of receiving inadequate information related to sexual healthcare, given that they may be perceived as “too young” to warrant these discussions [11,12,13,14,15,16,17,18,19,20,21,22,23,24,25,26,27,28,29,30]. As noted in the qualitative feedback, some younger participants felt that the video content was less relevant to their experiences, reinforcing the need for developmentally appropriate educational tools that reflect the diverse needs of the AYA population. Ensuring that younger adolescents are not excluded from these conversations due to perceived readiness is essential to promoting equitable access to information.

Importantly, our findings revealed that AYA wanted more guidance on how to talk with partners about sexual health. This reflects a broader gap in AYA cancer care, as recent studies have shown that partner communication is a common unmet need and that relationship-focused coping strategies can play an important role in managing sexual health during and after treatment [31,32]. This highlights a need for targeted supports that specifically address sexual health in the context of relationships, extending beyond general education on sexual health to include guidance on navigating intimacy and partner communication. Future revisions of the video will therefore include expanded content on partner communication, consent, and managing psychosocial impacts of treatment. These areas emerged as the most frequently reported needs and represented concrete targets for improvement in subsequent iterations of the tool.

Several limitations should be considered when interpreting our findings. We aimed for national representation across Canada; however, most participants were recruited from a single province (Ontario), which may limit the generalizability of results to other regions. The findings are likely relevant beyond Canada, as sexual health remains under-discussed in AYA Oncology care internationally [16]. Still, generalizability may be influenced by differences in healthcare systems, cultural attitudes toward sexual health, and political contexts that shape how openly these issues can be addressed [33]. Because participation required English proficiency, accessibility may have been limited for AYAs from other linguistic or cultural backgrounds. Future work should explore translating the material to increase inclusivity and equity, as well as partnering with community organizations to better distribute the resource beyond tertiary care centers. Additionally, because the video was available online, we cannot rule out the possibility that some participants had viewed it prior to study participation, potentially influencing baseline knowledge scores. As this was a pre-post design without a control group, improvements in knowledge could partly reflect testing effects or participant self-selection rather than the intervention alone. The questionnaire also required a significant time commitment (approximately 30 min), which may have resulted in selection bias by attracting more motivated or health-literate individuals. Finally, the video was developed with input from various healthcare providers, but patients were not directly involved in the design process. Future development of educational tools should incorporate patient partners to ensure the content fully reflects the lived experiences and priorities of the target population.

This study also has several strengths. It is, to our knowledge, the first to evaluate a whiteboard video focused specifically on sexual health education for female AYAs with cancer, a topic that remains underrepresented in oncology research and patient education. The pre-post mixed-methods design captured both knowledge change and participants’ impressions of the video’s clarity, relevance, and acceptability. The video was assessed using established tools for understandability and actionability, supporting the practicality and potential applicability of whiteboard videos as an educational approach within AYA oncology care.

The video is currently used in clinical practice within the PM Cancer Center AYA Program. During intake visits, a clinical nurse specialist identifies patients with sexual health concerns and shares the video as an educational resource to facilitate discussion. Depending on the patient’s needs, this may be followed by additional guidance or referral to specialized supports such as gynecology, therapy, or pelvic health services. While this represents an early stage of integration into care, future work should examine optimal timing, delivery methods, and follow-up approaches to support broader implementation across oncology settings.

## 5. Conclusions

In summary, our study demonstrates that a whiteboard education video is an effective tool for improving sexual health knowledge among AYA females with cancer, with participants rating it highly for clarity, practical value, and overall impact. Qualitative feedback further underscored the value of normalizing sexual health conversations, as well as the need for content that is concise and addresses both age- and relationship-specific factors. Informed by this feedback, we plan to revise the current video and develop two distinct tools: one tailored for older AYAs and another specifically designed for younger adolescents, ensuring developmental appropriateness and relevance across the AYA spectrum. This tailoring will involve adjusting the language, examples, and tone to reflect differing levels of maturity, life experience, and comfort discussing sexual health.

These findings suggest that whiteboard-style educational tools may be useful in supporting sensitive topics within and beyond cancer care. Their ability to deliver clear, engaging, and visually appealing messaging makes them particularly suited to AYA populations and adaptable across diverse topics. Future studies should explore the scalability of these tools in routine oncology care, assess their impact on communication outcomes and patient behavior, and refine them by tailoring content to different developmental stages, relationship contexts, and cultural differences. Examining the optimal timing of sexual health education, with opportunities to revisit the topic across the cancer trajectory, will also be essential to ensure these resources meet the diverse and evolving needs of AYA patients. Building on this work, subsequent phases will focus on piloting implementation within AYA programs and evaluating their feasibility and impact in real-world clinical settings.

## Figures and Tables

**Figure 1 curroncol-32-00681-f001:**
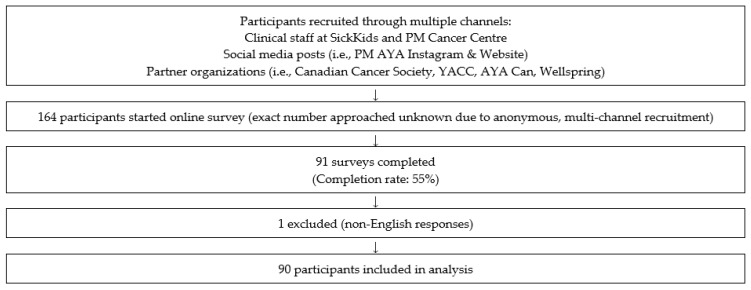
Participant Flow Diagram.

**Figure 2 curroncol-32-00681-f002:**
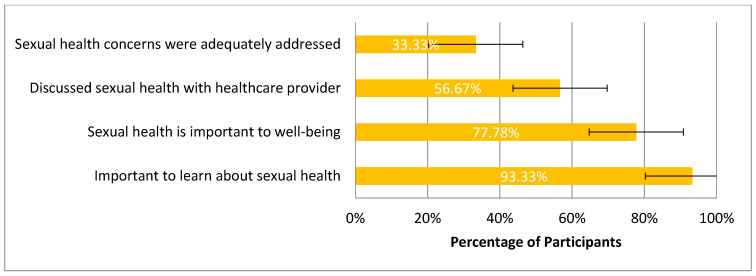
AYA Perspective on Sexual Health and Sexual Health Discussions.

**Figure 3 curroncol-32-00681-f003:**
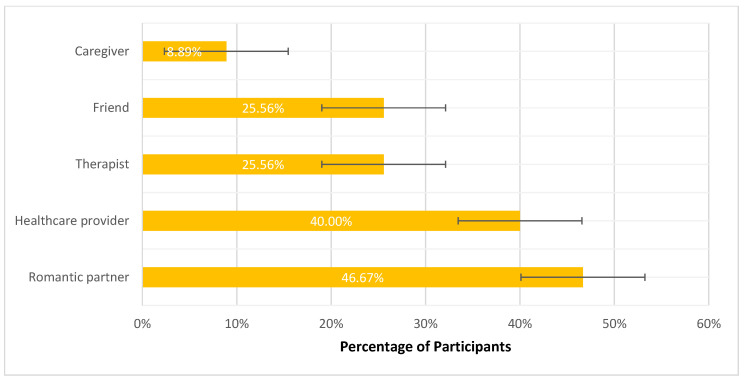
Who Participants Talked to About the Impact of Cancer on Sexual Health.

**Figure 4 curroncol-32-00681-f004:**
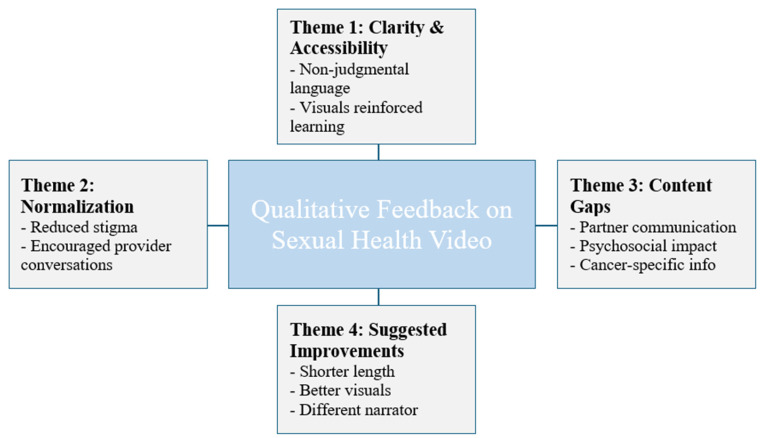
Themes from Qualitative Analysis.

**Table 1 curroncol-32-00681-t001:** Demographics and Medical Characteristics of Participants.

Category	N = 90	%
**Age**		
15–19	21	23.3%
20–29	21	23.3%
30–39	48	53.3%
**Gender Identity**		
Woman	88	97.8%
Transgender man	2	2.2%
**Race/Ethnicity**		
White	43	47.8%
Black	6	6.7%
Asian	15	16.7%
Arab	1	1.1%
South Asian	7	7.8%
Indigenous	1	1.1%
Other	15	16.7%
Prefer not to answer	2	2.2%
**Country of Birth**		
Canada	71	78.9%
Other	19	21.1%
**Current province**		
Ontario	80	88.9%
British Columbia	6	6.7%
Alberta	2	2.2%
Manitoba	1	1.1%
New Brunswick	1	1.1%
**Education**		
Public or grade school	23	25.6%
Completed high school	15	16.7%
Completed college or university	37	41.1%
Completed technical school	1	1.1%
Completed professional degree (Combined with post grad)	6	6.7%
Completed a post-graduate degree	17	18.9%
**Language spoken at home**		
English	73	81.1%
Other	17	18.9%
**Sexuality**		
Heterosexual	71	78.9%
Gay/Lesbian	2	2.2%
Bisexual	8	8.9%
Pansexual	3	3.3%
Asexual	1	1.1%
Other	2	2.2%
Prefer not to answer	3	3.3%
**Cancer type**		
Breast cancer	17	18.9%
Leukemia	14	15.6%
Lymphoma	10	11.1%
Bone tumor	9	10.0%
Soft tissue tumor	6	6.7%
Brain tumor	5	5.6%
Ovarian	7	7.8%
Colon cancer	4	4.4%
Germ cell tumor	3	3.3%
Cervical	3	3.3%
Other	12	13.3%
**Time from Treatment**		
Currently receiving treatment	50	55.6%
Less than 6 months post-treatment	6	6.7%
Between 6 months and 1 year post-treatment	11	12.2%
Between 1 year and 5 years post-treatment	17	18.9%
Between 5 years and 10 years post-treatment	6	6.7%

Participants could select more than one response for education level; therefore, percentages may sum to greater than 100%. “Other” refers to participant-reported identities or categories not represented in predefined response options. “Prefer not to answer” indicates participants who chose not to disclose this information. A small number of transgender men were included due to assigned female at birth eligibility criteria; their responses were analyzed in aggregate to support inclusion while protecting confidentiality, and subgroup analyses were not conducted due to small sample size.

**Table 2 curroncol-32-00681-t002:** Multiple Linear Regression Predicting Knowledge Score Change.

Predictor	B	SE	95% CI	*p*-Value
Intercept	34.50	9.68	15.26, 53.75	<0.001
Age (years)	−0.24	0.46	−1.16, 0.68	0.609
Treatment status (yes vs. no)	1.48	4.41	−7.29, 10.26	0.737
Education (ref: HS or less)	−17.26	8.27	−33.71, −0.81	0.040

## Data Availability

The original contributions presented in this study are included in the article/Appendix A. Further inquiries can be directed to the corresponding author.

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
