# Peer review of "Informed and Empowered: A Pre–Post Evaluation of a Whiteboard Video for Sexual Health Education in Female Adolescents and Young Adults with Cancer"

_curroncol, 2025, doi:10.3390/curroncol32120681_

Round 1

Reviewer 1 Report

Comments and Suggestions for Authors

Overall Comments

Thank you for the opportunity to review this timely and thoughtfully designed study that evaluates a 13-minute whiteboard video aimed at improving sexual health knowledge among female adolescents and young adults (AYAs) with cancer. The manuscript is clearly written and reports a pre- and post-intervention knowledge gain of approximately 19.5% among 90 participants, along with strong ratings for acceptability. The authors effectively frame their work within best practices for patient education, as evidenced by high PEMAT scores, and within the context of AYA oncology.

This topic is vital and often under-addressed in routine care. The authors’ plan to iteratively tailor materials based on developmental stages is both pragmatic and compelling. The considerations outlined below aim to enhance methodological transparency (including questionnaire development and psychometrics, handling of missing data, and analysis choices), improve clarity in reporting (such as participant flow and recruitment denominators), and refine the discussion regarding inclusivity, generalizability, and practical implementation.

Key strengths to retain:

  1. Precise problem framing and a clinically relevant intervention addressing a documented care gap.
  2. A concise description of the intervention development that is highly understandable and actionable (as per PEMAT-A/V), accompanied by detailed supplemental scoring.
  3. Statistically and practically meaningful knowledge gains, along with clinically relevant subgroup patterns.

Main Recommendations in Brief:

  1. Expand on the methods used in questionnaire design by including examples of items, details about piloting, and the rationale for scoring. Additionally, provide any available reliability metrics.
  2. Clarify the participant flow and recruitment sources by distinguishing between in-person and national outreach. Include specific numbers at each stage, as well as reasons for participant exclusion or decline. Figure 1 should be made more detailed.
  3. Report effect sizes, such as Cohen’s d, alongside p-values. Consider conducting sensitivity analyses, for instance, by excluding prior viewers if detectable.
  4. Address inclusivity in language and the analytical treatment of the small subgroup of transgender men who qualify based on being “assigned female at birth.”
  5. Discuss the logistics of implementation, including where and how the process fits within the workflow, staff training requirements, maintenance, and update frequency. Also, consider potential equity issues given the Ontario-centric sample.

Title

The title should be clear, informative, and aligned with the content. It may be beneficial to explicitly note the study design (e.g., “pre–post evaluation”) to set expectations for causal inference.

Abstract

The study is well structured and balanced. You report the primary outcome of knowledge improvement (↑19.5%, p<0.001) and high PEMAT scores. To enhance interpretability, consider adding the 95% confidence interval you report later (14–24%) and one key acceptability metric, such as 89% would recommend it.

Introduction

The rationale is clearly established, focusing on the unmet sexual health needs in adolescents and young adults (AYA) with cancer, as well as the appropriateness of using whiteboard videos for addressing sensitive topics. For clarity, consider explicitly stating the primary and secondary aims at the end of this section in a single sentence.

Methods

  1. Intervention Development & Measures: A thorough description of the multi-disciplinary process, along with PEMAT and readability assessments, is provided. Please include the following:

- Examples of the nine knowledge items, their alignment with the learning objectives, and any details on piloting; report item difficulty pre- and post-pilot if available.

- Clarify the readability reporting, specifying which calculator is primary, and justify the grade-level threshold used.

- Correct capitalization: it should be “REDCap,” not “RedCap.”

  1. Sampling & Recruitment: Provide counts by channel (clinic identification, social media, partner organizations) along with a CONSORT-style flow diagram that includes denominators, detailing how many were invited, how many started, and how many completed the study (noting that 164 started, 91 completed, and one was excluded). If permissible, add a table or appendix listing reasons for in-person declines.
  2. Ethics & Consent: The section is clear; consider adding the date(s) of REB approval and registration, if applicable.
  3. Statistical Analysis:

- Clearly report the a priori primary endpoint and whether any adjustments for multiple comparisons were necessary (for example, for several secondary analyses).

- Provide effect size (Cohen’s d for paired t-tests) and standardized β for regression, along with model diagnostics (such as residual plots and multicollinearity checks).

- Clarify the handling of missing data (e.g., “complete cases” or imputation) and specify the exact sample size for each analysis, noting any exclusions in the logistic regression.

Results

  1. Education Best Practices: The use of PEMAT is effective. Consider moving the most influential low-scoring actionability item from the supplement into the main text for better visibility to readers.
  2. Participant Characteristics: The table provided is helpful. Given the predominance of participants from Ontario, explicitly quantify and address potential regional bias in the text.
  3. Primary Outcome: The pre–post gain and 95% confidence interval are clear. Please add the effect size and include a sensitivity analysis, such as excluding participants who may have viewed the video previously, if identifiable.
  4. Subgroup/Predictors: The regression table is well-reported, showing smaller gains associated with higher education levels. The negative correlation with age in the bivariate analysis and the non-significant multivariable effect related to age warrant a brief explanatory note regarding issues such as collinearity and power.
  5. Communication & Satisfaction: The figures are useful; consider providing exact numerators and denominators in the captions, as well as confidence intervals for key proportions (e.g., 89% recommendation rate).

Figures

- Figure 1 (flow): Expand on recruitment channels and provide detailed reasons for attrition to enhance reproducibility.

- Figures 2–3: Include sample sizes (Ns) and confidence intervals (CIs) for the bars, and ensure consistent use of terminology (e.g., use "healthcare provider" instead of "medical team").

- Figure 4 (themes): Consider adding example quotes for each theme in the caption or supplementary materials.

Tables

Table 1: Well organized. Add footnotes for category definitions ("Other," "Prefer not to answer") and clarify the small subgroup of transgender men included due to AFAB eligibility. Consider an inclusivity note describing how their data were analyzed.

Discussion

The study's design is balanced and appropriately cautious given its pre–post framework. To strengthen the findings, consider the following suggestions:

  1. Discuss the limitations related to the Ontario-heavy sample and the requirement for English language proficiency. Propose strategies to enhance equity and accessibility, such as providing translations and localized dissemination methods.
  2. Provide more details about the implementation process. Specify where in the care pathway the video will be integrated, who will introduce it, and outline follow-up prompts or referral pathways for sexual health services.
  3. Clearly address the tailoring of the intervention for different developmental stages. It's great that you plan to create separate tools for younger and older adolescents and young adults (AYAs). Additionally, include support for partner communication.

Conclusion

The text is concise and aligned with the results. Next steps include co-designing with patient partners and piloting implementation.

References

The current citations are relevant; however, there are minor points to address:

  1. The Shoemaker PEMAT reference appears twice (items 13 and 17). Please deduplicate it and ensure consistent formatting throughout.
  2. When discussing readability grade-level recommendations, consider citing a specific authoritative guideline. The current form of your reference 18 is brief and could be expanded for clarity.

Suggestions for Minor Edits and Formatting:

  1. Standardize the capitalization of acronyms upon their first use (e.g., AYA, PEMAT-A/V, REDCap).
  2. Ensure consistent reporting of percentages along with their denominators, and include 95% confidence intervals (CIs) for key acceptability outcomes whenever possible.
  3. Consider condensing a few sentences in the Introduction to maintain a focused narrative before stating the aims.

Supplementary materials:

The PEMAT scoring sheet is helpful and supports the main text. Consider adding a brief legend to explain how the overall percentages were calculated, and explicitly connect the lowest-scoring actionability item to your proposed revisions.

*Recommendation*

Minor to moderate revisions are needed. This work addresses a significant gap and presents promising results. By including additional methodological details, improving the clarity of the flow reporting, and expanding the discussion on implementation and inclusivity, the manuscript will become stronger and more actionable for oncology programs looking to adopt similar tools.

Reviewer 2 Report

Comments and Suggestions for Authors

great work, a real pleasure to review.

see attached for the minor comments I had about choice of variables, simple summary and the addition of strengths

Reviewer 3 Report

Comments and Suggestions for Authors

        In the manuscript titled “Informed & Empowered: A Whiteboard Video for Sexual Health Education in Female Adolescents and Young Adults with Cancer,” the authors developed a 13‑minute whiteboard animation and demonstrated significant pre–post gains in sexual health knowledge among female AYA with cancer. This study contains interesting findings and is valuable for patient education and survivorship care in oncology. However, insufficient methodological transparency (evidence-to-content mapping) and limited detail on adaptation to identified content gaps are the major flaws of the study. Therefore, MAJOR revision has to be done before this manuscript could be accepted for publication in Current Oncology.

Major comments

  1. ”2.1. Intervention Development: Video Creation “—— The manuscript states that “the AYA Program at the PM Cancer Centre performed an environmental scan of available patient resources” and found no succinct, easy‑to‑understand tools. However, the scan methodology is not described. 
  2. ”2.1. Intervention Development: Video Creation “——The manuscript states that “The content of the video was grounded in current evidence-based information to 99
    ensure clinical accuracy and relevance" .Please provide explicit, itemized mapping from the video’s core messages to specific evidence sources (e.g., ASCO/ESMO/NCCN guidance, key systematic reviews/primary studies) 
  3. PEMAT-A/V——report rater backgrounds, training, inter-rater reliability (e.g., ICC/κ), and discrepancy resolution.
  4. Results 3.6 identify recurring needs—partner communication and consent, psychosocial impacts, treatment‑specific nuances, and so on. However, the Discussion do not translate these findings into a concrete improvement plan. Please add explicit, actionable revisions, and prioritize the highest‑demand areas.

Minor comments

  1. Figure 1 (Participant Flow Diagram): Add reasons for refusal/non-completion with counts : lack of interest, not fluent in English, discomfort with the topic, time constraints, other/unknown.
  2. Table 1 (Demographics and Medical Characteristics): Add subgroup analyses.
  3. Readability ranged from grade 10 to college level. Clarify whether scores refer to the full script, on-screen text, and/or subtitles.

Round 2

Reviewer 2 Report

Comments and Suggestions for Authors

thanks a lot for providing the additional details which certainly enhance the quality of the manuscript

no additional comments.

Reviewer 3 Report

Comments and Suggestions for Authors

In the manuscript titled “Informed & Empowered: A Pre-Post Evaluation of a Whiteboard Video for Sexual Health Education in Female Adolescents and Young Adults with Cancer,” the authors developed and evaluated a 13-minute whiteboard animation video aimed at improving sexual health knowledge among female AYAs with cancer. The study addresses an important and often underexplored aspect of survivorship care in oncology and highlights the potential of innovative educational tools to improve patient outcomes. While the findings are promising and the study holds value for advancing patient education, several significant issues remain unresolved. Consequently, major revisions are required before this manuscript can be considered for publication in Current Oncology.

Major Revision
The background section lacks a comprehensive literature review, which diminishes the reader's ability to fully grasp the urgency and severity of the current issues. Without a detailed review of existing tools, interventions, and gaps in the literature, the manuscript fails to highlight why this topic is both timely and necessary. Furthermore, the absence of a robust analysis of the limitations of existing approaches, such as pamphlets, traditional videos, or one-on-one consultations, weakens the justification for developing the whiteboard animation video.

Minor Revision
There are still many details in the manuscript that require attention. For example, in line 66, there is an error in the numbering of the reference citation.
